# A new synonym for *Viburnum erosum* (Viburnaceae) in East China, based on morphological and molecular evidence

**Liaocheng Zhao[1,2], Yitian Wang[2], Wenjun Lyu[3], Zijian Tang[2], Lihong Qiu[2,4]\*, Ming Tang[2,4]\***

**1** Laboratory of Systematic Evolution and Biogeography of Woody Plants, College of Ecology and Nature Conservation, Beijing Forestry University, Beijing, China, **2** College of Forestry, Jiangxi Agricultural University, Nanchang, Jiangxi, China, **3** National Germplasm Repository of Viburnum, Wuhan Botanical Garden, Chinese Academy of Sciences, Wuhan, Hubei, China, **4** Jiangxi Provincial Key Laboratory of Conservation Biology (2023SSY02081), Jiangxi Agricultural University, Nanchang, Jiangxi, China

\* tangming@jxau.edu.cn (MT); qiulihong@jxau.edu.cn (LQ)

## Abstract

The critical observations of living plants in the field, along with the examination of type specimens and protologues, led us to conclude that the key characteristics, including the length of the petiole, stipules and peduncles, as well as the shape of leaves of *Viburnum fengyangshanense*, all fall within the variation range of *V. erosum*. Additionally, molecular analysis of nuclear ribosomal internal transcribed spacer (nrITS) and three plastid DNA markers (*rbcL*, *matK* and *ndhF*) indicates that *V. fengyangshanense* and *V. erosum* are deeply nested within a clade. Therefore, based on morphological and molecular evidence, it is demonstrated that *V. fengyangshanense* should be regarded as a new synonym of *V. erosum*.

## Introduction

*Viburnum* Linnaues [1] comprises approximately two hundred species of shrubs and small trees worldwide, distributed from the subtropical northern hemisphere to the Andes Mountain and tropical Asia. China is one of the core distribution centers of this genus with 73 species (45 endemics) [2]. *Viburnum* was originally classified within Caprifoliaceae [3,4], phylogenetic studies based on morphological traits and molecular data indicate that *Viburnum* differs from other genera in Caprifoliaceae and has closer genetic relationship with Adoxaceae [5], which includes *Adoxa* Linnaeus [1], *Tetradoxa* C.Y. Wu [6], and *Sinadoxa* C.Y. Wu, Z.L. Wu & R.F. Huang [7]. This classification has been widely accepted in the taxonomic community [2,8,9]. However, some researchers have proposed that the scientific name of this family should be Viburnaceae. Reveal [10] suggested that if *Viburnum* was to be established as an independent family, the name Viburnaceae should be conserved, while Adoxaceae should be "superconserved" in the event of a merger between the two families. The International Code of Nomenclature (ICN) [11] approved the retention of Viburnaceae

**Data availability statement:** All relevant data are within the manuscript and its Supporting Information files.

**Funding:** The research was supported by the National Natural Science Foundation of China (Grant No. 31960043) and Key Technologies Research and Development Program (2024YFF1307400). The funder played a critical role in research design, data collection and analysis, decision to publish, and preparation of the manuscript.

**Competing interests:** The authors have declared that no competing interests exist.

but the proposal to conserve Adoxaceae was rejected. The adoption of these recommendations indicates that Viburnaceae should be recognized as the scientific name for a new taxa formed together with *Viburnum*, *Sambucus* Linnaeus, *Tetradoxa*, *Sinadoxa* and *Adoxa*.

*Viburnum erosum* Thunberg [12] was first collected by Carl Peter Thunberg during his botanical exploration in Japan, but the species was initially described and published by Johan Andreas Murray [12] based on *C. P. Thunberg s.n.* (UPS) (Fig 1A) from Japan without specifying a precise locality. Due to its considerable variability in various morphological features, such as petiole and stipule length, and particularly the density of indumentum on stems, abaxial leaves and peduncles [2,4,13,14], serveral names, including *V. erosum* var. *atratocarpum* Hsu [13] (Fig 2A), *V. erosum* var. *ichangense* Hemsley [15] (Fig 2B), *V. erosum* var. *setchuenense* Graebner [16] (Fig 2C), *V. taquetii* H. Léveillé [17] (Fig 2D) were initially published as new taxa but later considered as synonyms of *V. erosum* (Fig 1A) for their traits falling within the variation range of the latter [2,4,13]. In addition, *V. erosum* represents a taxonomic complex with significant morphological variations, and widely distributed across most provinces of China and other Asian regions, such as South Korea, North Korea and Japan [2,4,12,18].

*Viburnum fengyangshanense* Z.H. Chen, P.L. Chiu, et L.X. Ye [19] was first described based on *Z. H. Chen et al. LQ2016001* (holotype, ZM; isotype, HTC, ZM; color plate Fig 1B) collected in bush and slope nearby Fengyang Lake in Fengyang Mountain, Longquan County, Zhejiang, China. Qiu noted in the protologue that this species is distinct from other *Viburnum* species for its longer stipules and bigger leaves [19]. He also claimed that it resembles *V. erousm* in certain quantitative traits, such as larger leaves and fruits, and longer peduncles, stipules, petiole, but he did not provide type specimens of this species nor conduct molecular biology study.

During our filed work in May 2023 in Zhejiang, China, we discovered a unique *Viburnum* population in Fengyang Mountain, Longquan County. These plants are 1–2 m tall, with compound umbel-like cyme featuring a 1.5-3 cm peduncle, ovate-elliptic leaves, lateral veins 6–9(-12) that directly ending in teeth,few glands at the leaf base and two persistent stipules at the base. Initially, we identified these plants as *Viburnum erosum*; however, considering that these plants are collected from the type locality of *V. fengyangshanense*, we compared the protologue and color plates provided by Qiu et al. [19], and found that these distinctive morphological traits also corresponded to *V. fengyangshanense*. In order to find the key morphological evidence that better distinguishes these two similar species, we futher scrutinized the type specimens, protologue and our previous collections of *V. erosum*. We believe that the discrepancies [19] between *V. fengyangshanense* (Figs 1 and 3) and *V. erosum* (Figs 1,2 and 4) fall within the variation range of the latter. These research results further raise an issue regarding the taxonomic relationship between *V. fengyangshanense* and *V. erosum*. To indetify the relationship between them, morphological and molecular phylogenetic studies were conducted to clarify their taxonomic status.

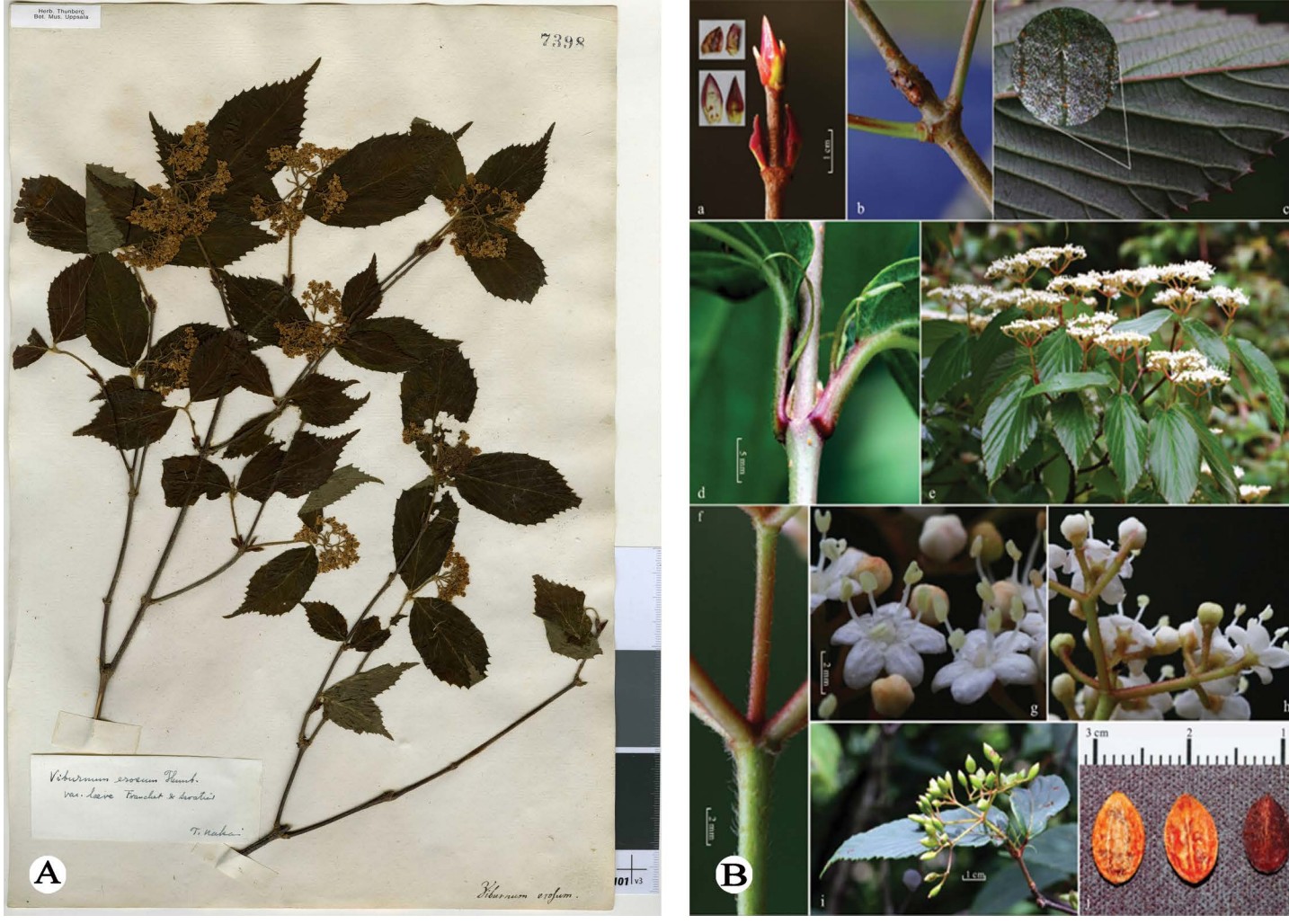

**Fig 1. *Viburnum erosum* and *V. fengyangshanense*.** (A) Holotype of *Viburnum erosum* [C. P. Thunberg s.n. (UPS), collected from Japan]. (B) Color illustration of *V. fengyangshanense* in protologue (Qiu et al. 2020).

## Materials and methods

### Morphological studies

We meticulously examined all the available high-resolution images of herbarium specimens of *Viburnum erosum* from A, AU, BNU, CCAU, CDBI, CSFI, CSH, FJFC, FKP, GFS, GNNU, GZTM, HNNU, HZ, IBK, IBSC, IFP, JIU, JXAU, K, KUN, L, NAS, NF, P, PE, QFNU, SDF, SZG, UPS, US, XBG, WCSBG and WUK. Since Qiu and Chen [16] did not provided any specimen images of *V. fengyangshanense*, we primarily refered to the color illustrations (Fig 1B) and morphological description in the protologue, and conducted careful observations of *V. fengyangshanense* collected in Fengyang Mountain, Longquan County, Zhejiang, China (Type locality). All major morphological characters of *V. erosum* and *V. fengyang-shanense* were studied on living plants using a digital camera (Olympus TG-6, Tokyo, Japan).

### Species sampling, DNA extraction and data collection

Based on previous phylogenetic studies of *Viburnum* [20–24], we selected 64 samples encompassing 61 species and 3 varieties across 11 sections within the genus. Following the taxonomic framework proposed by Donoghue et al. [20], we

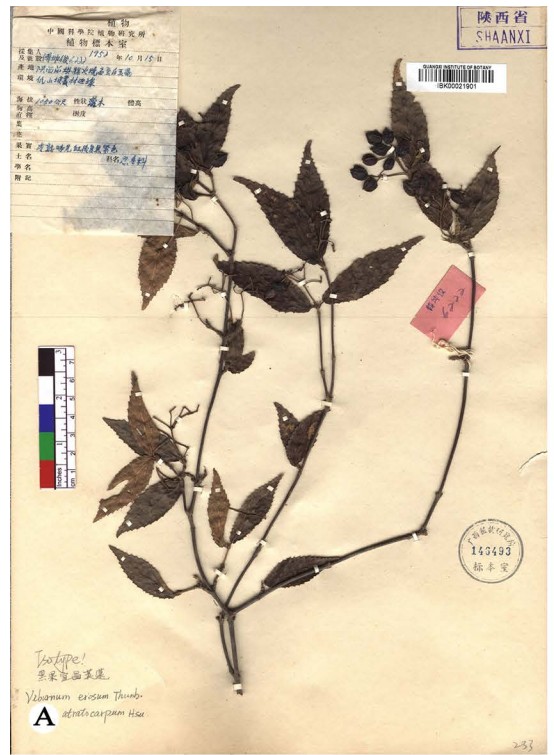

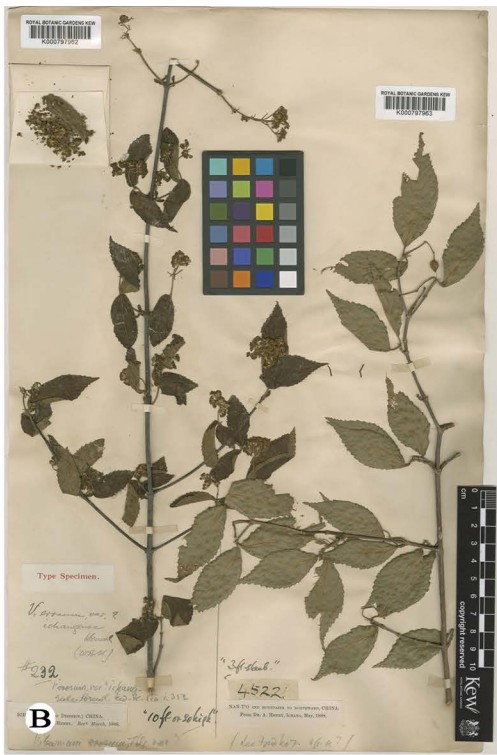

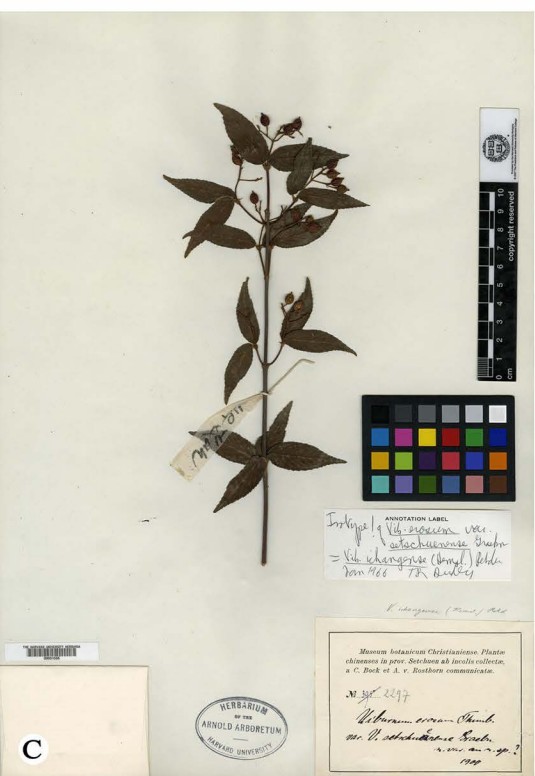

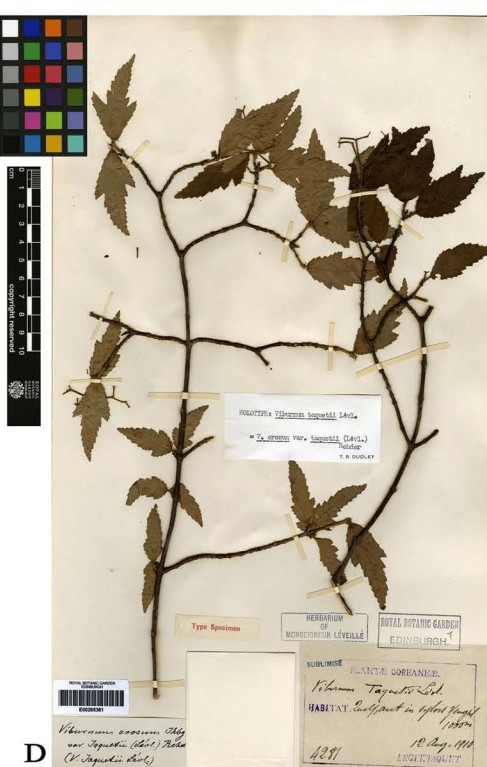

**Fig 2. *Viburnum erosum*.** (A) Huoshaodian Village, Liuba, Shaanxi, China, F. K. Jun 6222 (IBK, isotype of *V. erosum* var. *atratocarpum*). (B) Patung District from Ichang (Yichang), Hubei, China, A. Henry 232 (K, syntype of *V. erosum* var. *ichangense*). (C) Precise locality unkonwn, Sichuan, China, C. Bock & A. V. Rosthorn 2297 (A, isotype of *V. erosum* var. *setchuenense* Graebner). (D) Quelpaert in Sylvis Yengsil, South Korea, T. Taquet 4281 (A, holotype of *V. taquetii*).

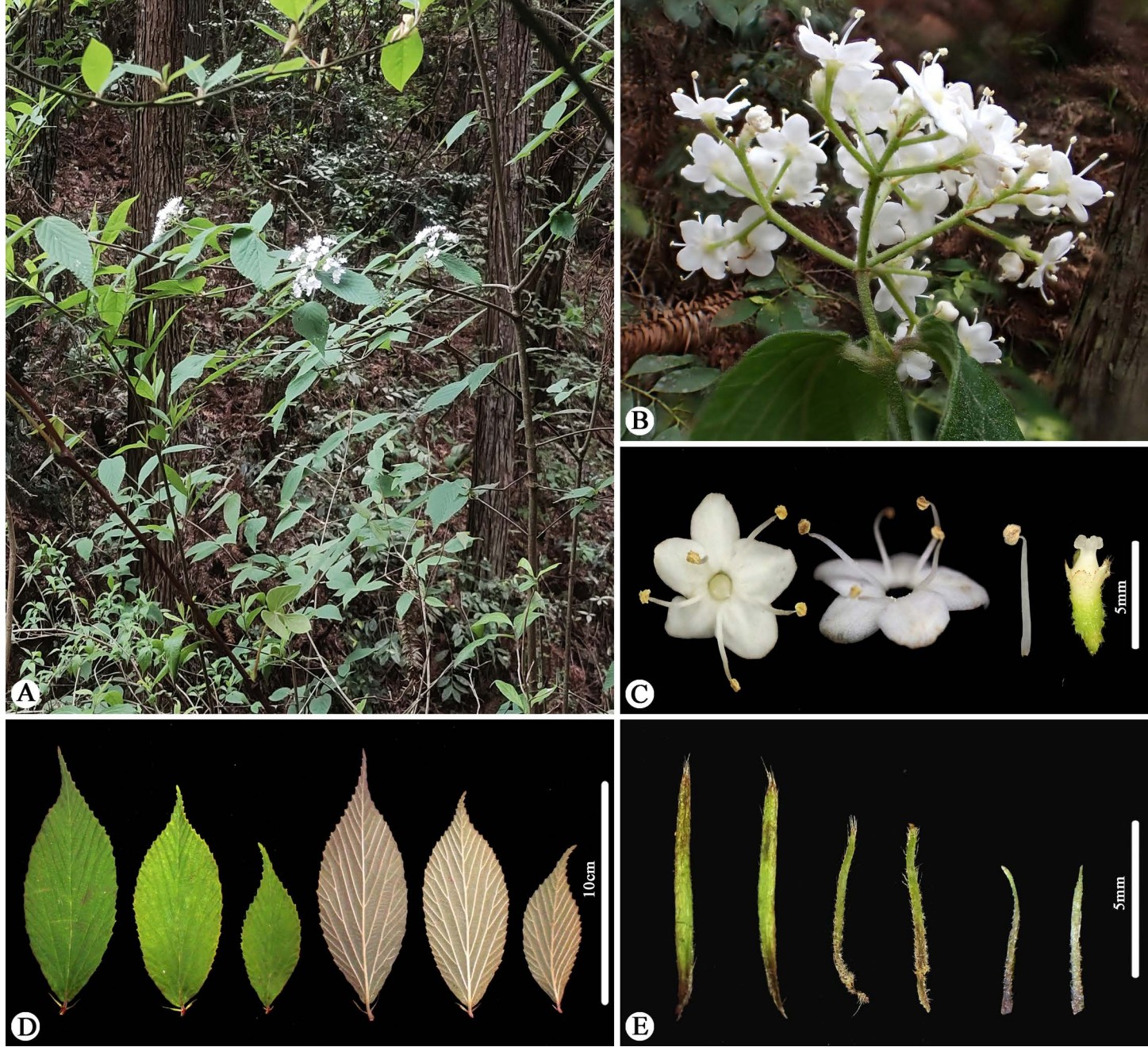

**Fig 3. Morphology of *Viburnum fengyangshanense* ( = *V. erosum*).** (A) Habitat. (B) Inflorescence. (C) Flowers. (D) Leaves. (E) Stipules.

designated *Sambucus canadensis* Linnaeus as the outgroup for our phylogenetic analysis. We generated new sequences of *V. erosum*, *V. fengyangshanense*, *V. lancifolium* P.S. Hsu and *V. squamulosum* P.S. Hsu, while sequences for other samples were obtained from the NCBI (https://www.ncbi.nlm.nih.gov/). Voucher information and GenBank accession numbers for the materials in this study are provided in S1 Table.

Total genomic DNA of *Viburnum erosum*, *V. fengyangshanense*, *V. lancifolium* and *V. squamulosum* were extracted from silica gel-dried leaves using the modified CTAB method [25]. DNA integrity was assessed by electrophoresis in a

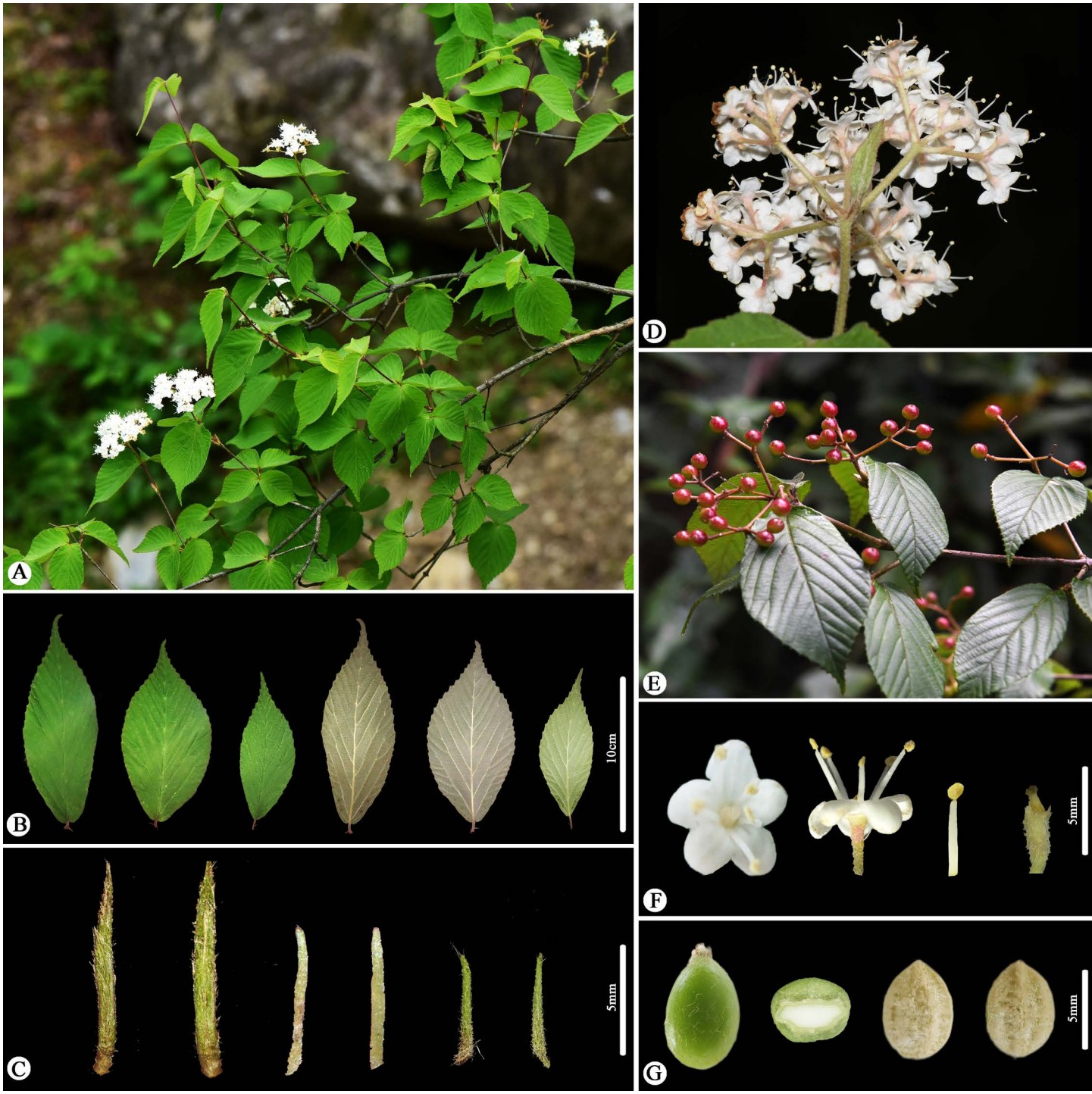

**Fig 4. Morphology of *Viburnum erosum*.** (A) Habitat. (B) Leaves. (C) Stipules. (D) Inflorescence. (E) Infructescence. (F) Flowers. (G) Fruit and seed.

1% (w/v) agarose gel, and DNA quality was evaluated using a NanoDrop spectrophotometer 2000 (Thermo Scientifc, Waltham, MA, USA). The DNA samples were subsequently sent to Novogene Bioinformatics Technology Co., Ltd. (Beijing, China) for library construction. Paired-end (2×150 bp) libraries were constructed using Nova-PE150 strategy, generating over 2 Gb of genome skimming data for each samples.

The raw skimming sequences were processed using Trimmomatic v. 0.39 [26]. to remove unpaired and low-depth reads, thereby enhancing the accuracy and quality of the assembly. The filtered data was assembled into the complete chloroplast (cp) genomes and nuclear ribosomal DNA (nrDNA) sequences with GetOrganelle v. 1.7.7 [27]. The assembly results were visualized and manually corrected using Bandage v. 5.6.0 [28]. The four complete cp genomes were annotated by CPGAVAS2 [29], with *Viburnum japonicum* (Thunb.) C. K. Spreng. (GenBank accession no. OP644292) as the reference sequence. Chloroplast makers (*matK, ndhF, rbcL*) were extracted from the assembled cp genomes using Genetic Prime 2020 [30], and nrITS was extracted from nrDNA using ITSx [31].

### Phylogenetic analysis

The matrices of all sequences were aligned with MAFFT 7.450 [32]. Maximum likelihood (ML) analysis was performed using IQ-TREE [33], and bootstrap analysis was conducted with 20 000 ultrafast bootstraps. The best-fit BIC model was determined by ModelFinder [34]. Bayesian inference (BI) was carried out by MrBayes 3.2.6 [35] with 3 000 000 generations, sampling every 1 000 generations to ensure the convergence (average deviation of split frequencies less than 0.01 and the effective sample sizes over 200). The first 25% of the sampled data were discarded as burn-in, and the remaining trees were used to estimate the posterior probabilities (PP). Bootstrap percentage (MLBS) values ≥ 70 [36] and PP values ≥ 0.90 [37] were considered strong support. Finally, the phylogenetic trees were visualized by Figtree1.4 (http://tree.bio.ed.ac.uk/software/figtree/).

## Results and discussion

### Morphological study

As illustrated in Figs 1-4, the indumentum on abaxial leaves and inflorescence, the shape and size of leaves, as well as the number of lateral veins in *Viburnum erosum*, exhibits considerable intraspecific variation. Qiu and Chen [19] pointed out that the differences in leaf size, stipule length, indumentum density on abaxial leaves and seed characteristics between *V. erosum* and *V. fengyangshanense* are crucial for their differentiation. However, based on prior descriptions [2,4] and our morphological research findings (Figs 1-4), there is an excessive variation in these quantitative traits, which is insufficient to distinguishing these two species, especially the seed trait, which is a significant criterion for the morphological classification of *Viburnum* plants [38,39], yet both species possess 2 shallow dorsal grooves and 3 shallow ventral grooves (Figs 1B and 4G). Qiu and Chen [19] indicated that *V. fengyangshanense* has 5–7 lateral veins and two pairs of bud scales (a short inner layer and a long outer layer). However, the number of leaf veins in this species should be 6–11 pairs (Figs 1B and 3), which falls within the variation range of *V. erosum* (6–14) [2,4]. Further more, as depicted in Fig 1A, two pairs of unequal bud scales can be clearly visible. Additionally, we also observed a few circular glands on both sides of the base on abaxial leaf (Figs 3 and 4), which serve as an important taxonomic basis in *Viburnum* [23].

### Phylogenetic study

Both ML and BI analyses have generated nearly identical topologies, with the ML tree presented herein as the primary phylogenetic framework (Fig 5). The concatenated analysis of nrITS, *matK*, *rbcL*, and *ndhF* sequences from 64 accessions robustly supported the monophyly of *Viburnum* (BS = 100, PP = 1.00). Within the genus, sect. *Odontotinus* was resolved as a strongly supported monophyletic group (BS = 99, PP = 1.00), occupying a phylogenetic position consistent with prior phylogenetic studies [20–24]. Notably, *V. fengyangshanense* formed a well-supported clade (BS = 95, PP = 1.00) with two accessions of *V. erosum*, further corroborating their close phylogenetic affinity.

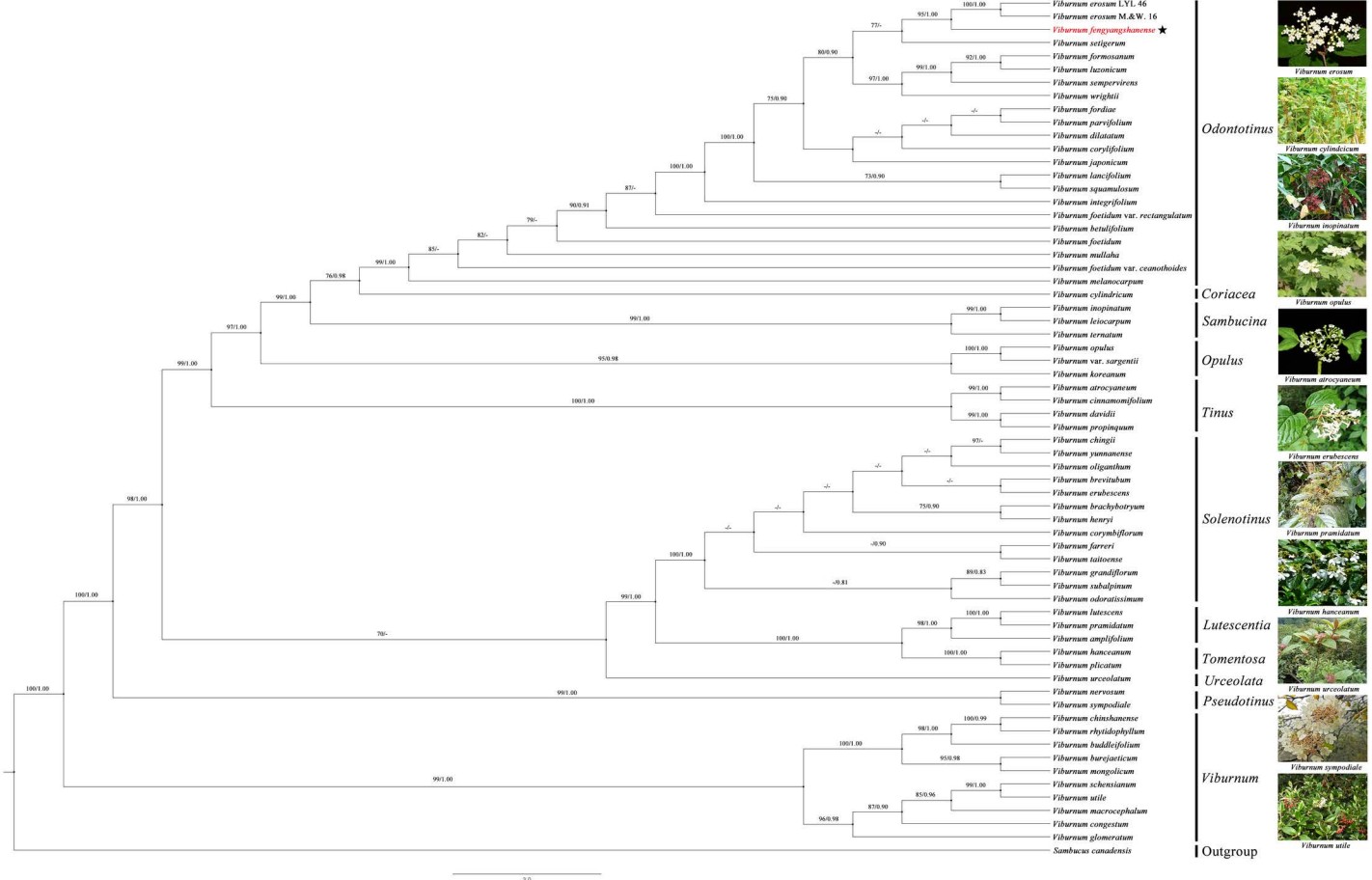

**Fig 5. The maximum likelihood tree of *Viburnum* based on nrITS, *rbcL*, *matK* and *ndhF*, with *V. fengyangshanense* highlighted in red font.** Bootstrap values (BS) and posterior probabilities (PP) are shown above the branches. Dashes (-) indicate MLBS < 70 or PP < 0.90.

In summary, based on the morphological and phylogenetic results of this study, we propose that *Viburnum fengyangshanense* should be regarded as a new synonym of *V. erosum* rather than a new species.

## Conservation implications

The taxonomic clarification of *Viburnum fengyangshanense* as a synonym of *V. erosum* has significant implications for biodiversity conservation in East China. First, this reclassification reduces taxonomic redundancy and refocuses conservation priorities by consolidating previously presumed distinct taxa into a single, morphologically and genetically variable species. This finding suggests that populations once considered endemic or narrowly distributed (e.g., *V. fengyangshanense*) may represent regional variants of a more widespread species. Consequently, conservation assessments should be updated to reflect this unified taxonomic framework, preventing the misallocation of limited resources toward redundant "species" and instead directing efforts toward safeguarding genetically diverse populations or ecologically critical habitats of *V. erosum*.

Furthermore, the recognition of broader intraspecific variation in *V. erosum* highlights the importance of conserving populations across its expanded geographic range, particularly in fragmented or threatened habitats. Protecting these populations ensures the maintenance of genetic diversity and adaptive potential, which is vital for the species resilience to

environmental changes. Additionally, this study emphasizes the necessity of integrating taxonomic revisions into conservation planning, as misidentification or oversplitting of taxa can lead to ineffective or counterproductive management strategies. By aligning conservation policies with robust taxonomic and molecular evidence, stakeholders can prioritize actions that address genuine ecological risks and promote the long-term survival of this ecologically significant *Viburnum* species in East Asia.

## Taxonomic treatment

***Viburnum erosum*** Thunberg [12]; Thunberg [18], (Figs 1A and 4).

　　Type:—JAPAN. Without precise locality, collection date unknown, *C. P. Thunberg s.n.* (holotype: UPS-V007398!) (Fig 1A).

　　=V. *fengyangshanense* Z.H. Chen, P.L. Chiu, et L.X. Ye [19], ***syn. nov.*** (Figs 1B and 3).

　　Type:—CHINA. Zhejiang, Longquan, Fengyang Moutain, Fengyang Lake, bush and slope, 1570 m a.s.l., 8 May 2016, *Z. H. Chen, L. X. Ye & S. L. Liu LQ2016001* (holotype: ZM; isotype: HTZ, ZM).

=*V. erosum* var. *atratocarpum* Hsu [13].

Type:—CHINA. Shanxi, Liuba, Huoshaodian Village, slope and forest margins, 1050 m a.s.l., 15 October 1952, F. K. Jun 6222 (isotype: IBK00021901!) (Fig 2A).

=*V. erosum* var. *ichangense* Hemsley [15].

Type:—CHINA. Hubei, Ichang (Yichang) Patung District, March 1886, A. Henry 232 (syntype: K000797962!) (Fig 2B); February 1887, A. Henry 2841 (syntype: K000797964!); Hubei, Ichang February 1887, A. Henry 1888 (syntype: K000797965!).

=V. erosum var. setchuenense Graebner [17].

Type:—CHINA. Sichuan, without precise locality, 1900, C. Bock & A. V. Rosthorn 2297 (isotype: A00031556!) (Fig 2C).

=*V. taquetii* H. Léveillé [17].

Type:—SOUTH KOREA. Sylvis Yengsil, Quelpaert, 1000 m a.s.l., 12 August 1910, T. Taquet 4281 (holotype: E00265361!; isotype: A00031557!) (Fig 2D).

　　The detailed morphological description refers to Hsu [4], and Yang & Malécot [2].

**Distribution and habit:**—*Viburnum erosum* is widely distributed in northern, southern, and southeastern China, as well as Japan, and Korea (Fig 6). It grows in forests and shrubs at elevations of 300−2300 m.

**Phenology:**—Flowering April to May; Fruiting August to October.

**Additional specimens examined:**—**CHINA. Anhui,** Huoshan, *M. B. Deng & Z. G. Wei 80166* (NAS00266165); Jinzhai, *K. Yao 9901* (NAS00614827); Qianshan, *X. H. Shi 39* (NF2015928); Shitai, *R. P. Jiang et al. 940* (NAS00614815); Yuexi, *Z. W. Xie & L. Zheng 97298* (PE02115525). **Chongqing,** *L. Q. Gao s.n.* (JXAU). **Fujian,** Longhai, C. M. Tan 971606 (SZG00044688); Nanping, S. S. You 989 (FJFC0002625); Zhangzhou, Nanjing, *T. P. Zhu 1233* (AU066161). **Gansu,** Chengxian, *Z. P. Wei 2227* (WUK0108949); Kangxian, *Y. Q. He & C. L. Tang 223* (WUK0215012); Wenxian, *Z. X. Hu 689* (IBSC0542031). **Guangdong,** Heping, *G. C. Zhang 96* (IBSC0541699); Renhua, *L. Tang 7443* (IBSC0541726); Shixin, *L. Tang 7132* (IBSC0541725); Wengyuan *X. Q. Liu 23929* (IBK00020993). **Guangxi,** Guanyang, *Z. Z. Chen 52136* (IBSC0541731); Pinggui, *Y. Liu et al. H0174* (IBK00323438); Quanzhou, *Z. S. Chung 81570* (IBSC0541730); Rongshui, *S. Q. Chen 16447* (PE0541728); Xingan, *L. G. Zhao & B. Lu 63203,* (IBK00189025). **Guizhou,** Dafang, *R. B. Jiang 705* (IBSC0542019); Leishan, *Y. F. Li 522634150830007LY* (GZTM0063374); Wengan, *J. B. Zuo D-370* (GFS0013538); Xingren, *Y. T. Zhang 7341* (IBSC0542021). **Henan,** Shangcheng, *C. S. Zhu et al. 13080835* (AU067206); Xinyang, *Plant Resource Investigation Team D0728* (PE01130026). **Hubei,** Baokang, *C. Q. Wang 3–001* (CCAU0007423); Hefeng, *H 1743* (CCAU0007856); Xianfeng, *W. B. Lin 512* (WUK0310836); Xuanen, *H. C. Li 3839* (IBSC0541876); Zhuxi, *P. Y. Li 9271* (WUK0356317). **Hunan,** Cili, G. B. Zhu 43 (IBSC0541840); Hengshan, H. T. Chang 3040 (IBSC0541864); Linwu, *L. Ang LA0144* (CSFI064335); Luxi*, D. G. Zhang 4331221607170767LY* (JIU31158);

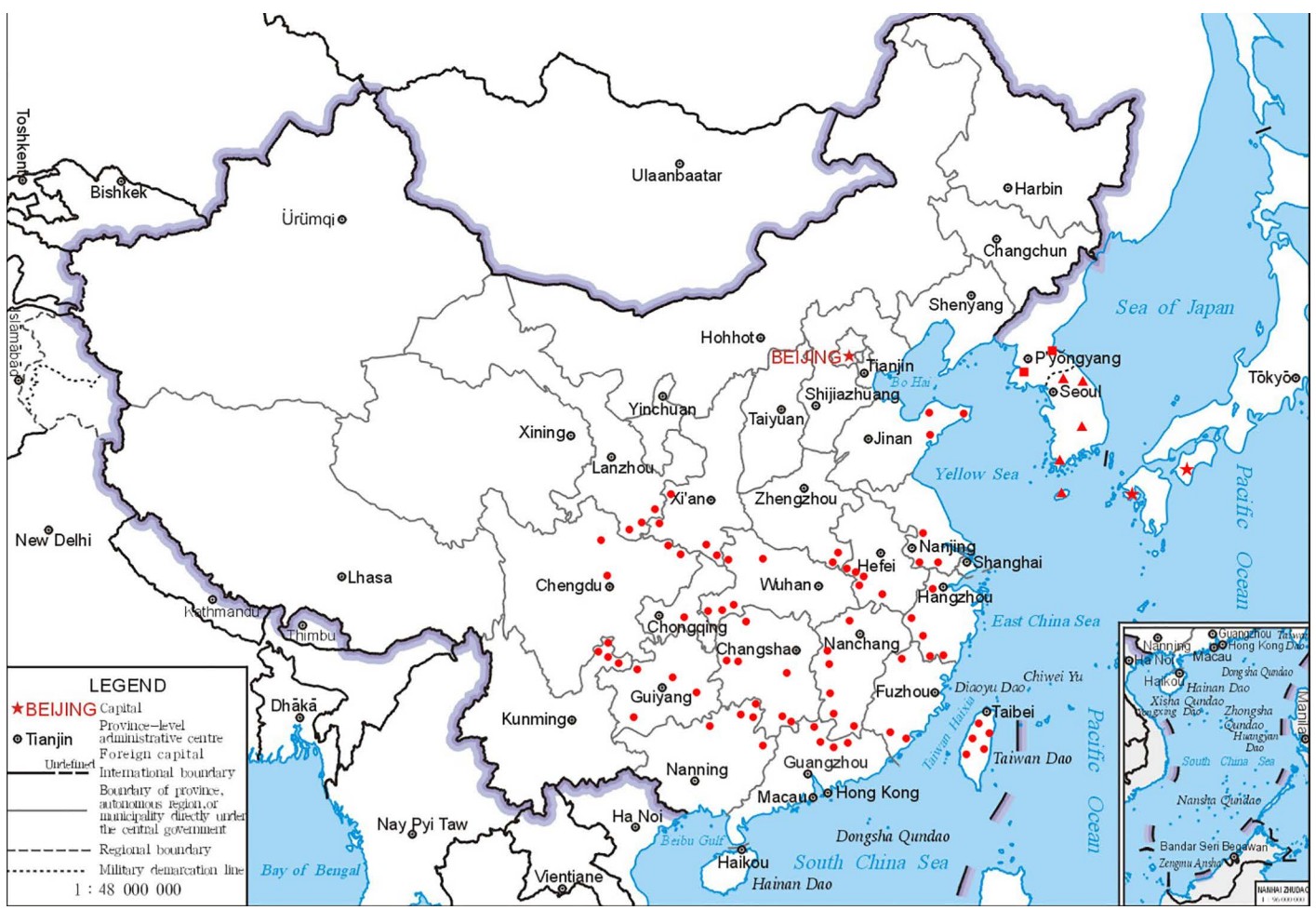

**Fig 6. The geographical distribution of *Viburnum erosum* is illustrated as follows: red dots indicate its distribution in China; the red stars represent its distribution in Japan; red rectangles indicate its distribution in North Korea; and red triangles represent its distribution in South Korea.**

Yizhang, *W. J. Liu & X. F. Song* (CSFI075586); Yuanling, *G. C. Zhang 360* (PE01130597). **Jiangsu,** Danyang, *Nanlin Team 7250* (IBSC0541990); Wuxi, *G. Yao et al. 10687* (NAS00138216); Yixing, *G. Yao 7071* (NAS00138215). **Jiangxi,** Chongyi, *Y. Y. Cong 33398* (HNNU00062513); Jinggangshan, *Y. L. Liu 46* (JXAU); Jiujiang, *Y. L. Liu 31* (JXAU); Xunwu, *Z. B. Tang* (GNNU0006512); Yichun, *H. G. Ye & F. Y. Zeng LXP 10–2435* (IBSC0775645), *X. H. Wu et al. 2* (JXAU). **Shandong,** Maoping, *L. Q. Qiu 4211* (WUK0312333); Qingdao, *T. Y. Zhou 1188* (NAS00266265); Yantai, *Y. T. Hou & C. Y. Guo 2014028* (QFNU0015829). **Shannxi,** Langao, *P. Y. Li 8470* (WUK0355776); Lueyang, *K. J. Fu 6016* (IBK00334551); Taibai, *K. J. Fu 5066* (WUK0059531); Zhenping, *G. Y. Xu 4910* (WUK0475774). **Sichuan,** Dayi, *H. Hai, Y. K. Li & Y. P. Du DY0469* (CDBI028543); Maoxian, *X. Y. Xiong et al. 2004005* (WCSBG007033); Nanjiang, *Y. He & H. Tian GWS20154161* (BNU0019539); Tongjiang, *Bashan Collection Team 5862* (PE02066229); **Taiwan**, Hualien, *R. Q. Gao et al. 1256* (KUN760743); Nantou, *J. G. Liu 995* (PE01130764); Taichung, *M. J. Liu 12* (PE01646064), *T. C. Chen 11881* (PE01775364), *11880* (PE00869145), *12116* (PE00869146); Taipei, Datong, *T. Y. Liu 933* (PE01446679); Yilan, *S. Y. Lv s.n.* (PE01130768), *Z. Y. Wang & J. G. Liu 43* (PE01130767, PE01130766). **Yunnan,** Shuifu, *W. G. Yang DT6662* (KUN15130048); Suijiang, *Z. H. Hu 1366* (IBSC0541733); Yiliang, *Z. Z. Ding & J. X. Wang 1091* (NAS00266608);

Zhenxiong, *S. Y. Bao 132* (KUN85618). **Zhejiang,** Anji, *H. X. Yu 24232* (HZ054851); Kaihua, *H. X. Yu 29711* (IBSC0541965); Qingyuan, *S. Y. Zhang 3472* (NAS00266285); Suichang, *Q. F. Zhang et al. ZX00213* (CSH0121231); Taishun, *S. Y. Zhang 3735* (HZ055065).

 **JAPAN. Nagasaki**, precise locality unknown, *C. L. Maximowicz s.n.* (L2974728, L2974729), (US 97369, US 97370); *R. Oldham 477* (L2974727); **Yokoska**, Nakatosa town, *T. Miyazaki 1405054* (P01168582); precise locality unknown, *H. Savatier 536* (US 97373), *M. Labbé 70* (P03318885), *363* (P03318884), *4154* (P03318879), *6185* (P03318880), *M. Labbé s.n.* (P03318881).

 **NORTH KOREA. Hwanghae-namdo**, precise locality unknown, *U. J. Faurie 680* (FKP86519); **Northern Gangwon-do**, precise locality unknown, *S. E. Liu 9398* (IFP13606999y0017), *G. Koidzumi s.n.* (FKP23836), *E. H. Wilson 10401* (FKP78065).

 **SOUTH KOREA. Gangwon-do**, precise locality unknown, *K. Kondo 9113* (PE01618021); Yangyang-gin, *K. Y. Chul KYC2007* (KUN1224955); **Chol-la-nam-do**, precise locality unknown, *W. H. Takahashi & C. S. Chang 344* (PE01618015), *378* (PE01618014), *C. S. Chang 1167* (PE01618016); **Incheon**, Ganghwa, *H. R. Na, K. Yoo & H. S. Kim N-EX11* (PE02112636); **Jeju-do**, Seogwipo, Jungmun, *B. U. Oh et al. 120612–006* (PE01920990); **Jeollanam-do**, Gangjin, *H. S. Jung 201128–0046* (PE02040482).

## Supporting information

**S1 Table. Voucher information and GenBank accession numbers for the materials in this study.** Star symbol indicates the newly obtained sequences.
(XLSX)

## Acknowledgements

We are grateful to the curators of the herbaria of A, AU, BNU, CCAU, CDBI, CSFI, CSH, FJFC, FKP, GFS, GNNU, GZTM, HNNU, HZ, IBK, IBSC, IFP, JIU, JXAU, K, KUN, L, NAS, NF, P, PE, QFNU, SDF, SZG, UPS, US, XBG, WCSBG and WUK for allowing us to examine and use their scanned images of specimens, especially for UPS providing the holotype of *Viburnum erosum*, where further promote our research. In addition, we would like to express gratitude to researcher Ren Chen of the South China National Botanical Garden of the Chinese Academy of Sciences for his constructive guidance.

## Author contributions

**Data curation:** Liaocheng Zhao, Yitian Wang.

**Formal analysis:** Liaocheng Zhao.

**Funding acquisition:** Ming Tang.

**Investigation:** Liaocheng Zhao, Yitian Wang, Wenjun Lyu, Zijian Tang, Lihong Qiu.

**Methodology:** Liaocheng Zhao, Ming Tang, Wenjun Lyu.

**Project administration:** Ming Tang.

**Resources:** Ming Tang.

**Software:** Liaocheng Zhao.

**Supervision:** Ming Tang.

**Visualization:** Liaocheng Zhao.

**Writing – original draft:** Liaocheng Zhao.

**Writing – review & editing:** Ming Tang, Lihong Qiu.

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
