## [Decision Letter · Decision Letter 0]

27 Jan 2025

PONE-D-24-45964A new synonym for Viburnum erosum (Viburnaceae) in East China, based on morphological and molecular evidencePLOS ONE

Dear Dr. Tang,

Thank you for submitting your manuscript to PLOS ONE. After careful consideration, we feel that it has merit but does not fully meet PLOS ONE’s publication criteria as it currently stands. Therefore, we invite you to submit a revised version of the manuscript that addresses the points raised during the review process.

We look forward to receiving your revised manuscript.

Kind regards,

Alia Ahmed

Academic Editor

PLOS ONE

Journal requirements: When submitting your revision, we need you to address these additional requirements. 1. Please ensure that your manuscript meets PLOS ONE's style requirements, including those for file naming. The PLOS ONE style templates can be found at https://journals.plos.org/plosone/s/file?id=wjVg/PLOSOne_formatting_sample_main_body.pdf and https://journals.plos.org/plosone/s/file?id=ba62/PLOSOne_formatting_sample_title_authors_affiliations.pdf. 2. Thank you for stating the following financial disclosure:  [National Natural Science Foundation of China (grant no. 31960043)].  Please state what role the funders took in the study.  If the funders had no role, please state: ""The funders had no role in study design, data collection and analysis, decision to publish, or preparation of the manuscript."" If this statement is not correct you must amend it as needed. Please include this amended Role of Funder statement in your cover letter; we will change the online submission form on your behalf. 3. Thank you for stating the following in the Acknowledgments Section of your manuscript: [We are grateful to the curators of the herbaria of A, AU, BNU, CCAU, CDBI, CSFI, CSH, FJFC, FKP, GFS, GNNU, GZTM, HNNU, HZ, IBK, IBSC, IFP, JIU, JXAU, K, KUN, L, NAS, NF, P, PE, QFNU, SDF, SZG, UPS, US, XBG, WCSBG and WUK for allowing us to examine and use their scanned images of specimens, especially for UPS providing the holotype of Viburnum erosum, where further promote our research. In addition, we would like to express gratitude to researcher Ren Chen of the South China National Botanical Garden of the Chinese Academy of Sciences for his constructive guidance. This work was supported by the National Natural Science Foundation of China (grant no. 31960043).]We note that you have provided funding information that is not currently declared in your Funding Statement. However, funding information should not appear in the Acknowledgments section or other areas of your manuscript. We will only publish funding information present in the Funding Statement section of the online submission form. Please remove any funding-related text from the manuscript and let us know how you would like to update your Funding Statement. Currently, your Funding Statement reads as follows:  [National Natural Science Foundation of China (grant no. 31960043)].   Please include your amended statements within your cover letter; we will change the online submission form on your behalf. 4. We note that your Data Availability Statement is currently as follows: [All relevant data are within the manuscript and its Supporting Information files.] Please confirm at this time whether or not your submission contains all raw data required to replicate the results of your study. Authors must share the “minimal data set” for their submission. PLOS defines the minimal data set to consist of the data required to replicate all study findings reported in the article, as well as related metadata and methods (https://journals.plos.org/plosone/s/data-availability#loc-minimal-data-set-definition). For example, authors should submit the following data: - The values behind the means, standard deviations and other measures reported;- The values used to build graphs;- The points extracted from images for analysis. Authors do not need to submit their entire data set if only a portion of the data was used in the reported study. If your submission does not contain these data, please either upload them as Supporting Information files or deposit them to a stable, public repository and provide us with the relevant URLs, DOIs, or accession numbers. For a list of recommended repositories, please see https://journals.plos.org/plosone/s/recommended-repositories. If there are ethical or legal restrictions on sharing a de-identified data set, please explain them in detail (e.g., data contain potentially sensitive information, data are owned by a third-party organization, etc.) and who has imposed them (e.g., an ethics committee). Please also provide contact information for a data access committee, ethics committee, or other institutional body to which data requests may be sent. If data are owned by a third party, please indicate how others may request data access. 5. Please include a caption for figure 1, 2, 3, 4, 5, and 6. 6. Please upload a copy of S1 Table to which you refer in your text on page 9. Please amend the file type to 'Supporting Information'. If the Supplementary file is no longer to be included as part of the submission please remove all reference to it within the text.

Reviewers' comments:

Reviewer's Responses to Questions

**Comments to the Author**

1. Is the manuscript technically sound, and do the data support the conclusions?

Reviewer #1: Yes

Reviewer #2: Yes

2. Has the statistical analysis been performed appropriately and rigorously? 

Reviewer #1: Yes

Reviewer #2: N/A

3. Have the authors made all data underlying the findings in their manuscript fully available?

Reviewer #1: Yes

Reviewer #2: Yes

4. Is the manuscript presented in an intelligible fashion and written in standard English?

Reviewer #1: Yes

Reviewer #2: Yes

5. Review Comments to the Author

Reviewer #1: This is an interesting contribution to the taxonomy of Viburnaceae and represents a proposal for synonymization supported by objective tools. Some comments and corrections have been included in the file in order to make the manuscript clearer.

Reviewer #2: Strengths:

Integration of molecular and morphological data.

Well-structured manuscript with clear figures.

Valuable contribution to Viburnum taxonomy.

Weaknesses:

Limited molecular sampling reduces the generality of conclusions.

Insufficient discussion on ecological and conservation implications.

Methodological details, especially bioinformatics, need clarification.

Limited comparison with prior phylogenetic studies.

6. PLOS authors have the option to publish the peer review history of their article (what does this mean? ). If published, this will include your full peer review and any attached files.

**Do you want your identity to be public for this peer review?** For information about this choice, including consent withdrawal, please see our Privacy Policy .

Reviewer #1: No

Reviewer #2: No

---

## [Author Response · Author response to Decision Letter 1]

18 Feb 2025

15 February 2025

Dear Reviewers,

We sincerely appreciate your insightful comments and constructive suggestions, which have significantly improved the quality of our manuscript. We have carefully addressed all the points raised in your review, and the revised manuscript now incorporates these changes. A clean version of the manuscript, along with a version highlighting all modifications (in red), has been uploaded for your convenience. Below, we provide a point-by-point response to your comments, detailing the specific revisions made.

Once again, we thank you for your time and expertise in evaluating our work. We believe the manuscript has been substantially strengthened through this revision process, and we look forward to your further feedback.

Yours sincerely,

Ming Tang

Response to Anonymous Reviewer 1:

Comment 1: Addition of references for Adoxa, Tetradoxa and Sinadoxa.

Response:

References for Adoxa Linnaeus [6], Tetradoxa C.Y. Wu [7], and Sinadoxa C.Y. Wu, Z.L. Wu & R.F. Huang [8] have been integrated into the reference section.

Comment 2: Clarification on the nomenclature of Viburnaceae.

Response:

The taxonomic rationale for retaining Viburnaceae has been revised to align with the International Code of Nomenclature (ICN) guidelines (Lines 42–45). Specifically, we emphasize that Viburnaceae was conserved under ICN Article 14.10, whereas the proposal to "superconserve" Adoxaceae was rejected, as per the rulings of the Nomenclature Committee for Vascular Plants.

Comment 3: Are such characteristics: leaf size, stipule length, indumentum density on abaxial leaves and seed, commonly used to recognize species in this genera?

Response: These morphological features are widely recognized as critical taxonomic criteria for identify Viburnum sect. Odontotinus [2,4], with both V. erosum and V. fengyangshanense being taxonomically placed within this section.

Response to Anonymous Reviewer 2:

Weakness 1: Limited molecular sampling reduces the generality of conclusions.

Response: In this study, we conducted comprehensive sampling of 64 samples representing 61 species and 3 varieties across 11 taxonomic sections within Viburnum, encompassing 83.4% (73 species) of the total Viburnum species distributed in China. Notably, sect. Odontotinus was particularly well-represented, with 19 species and 2 varieties sampled, accounting for 86.4% (22 species) of its Chinese distribution. This extensive sampling strategy provides a robust framework for reconstructing the phylogenetic relationships of Chinese Viburnum and offers critical evidence for resolving the phylogenetic placements of V. erosum and V. fengyangshanense.

Weakness 2: Insufficient discussion on ecological and conservation implications.

Response: A new subsection “Conservation Implications” has been added (Line 175-197 in “Revised Manuscript with Track Changes.docx”)

Weakness 3: Methodological details, especially bioinformatics, need clarification.

Response: The Materials and Methods section now details:

In this study, we employed second-generation high-throughput sequencing to generate complete chloroplast genomes and nuclear ribosomal DNA (nrDNA) assemblies for four Viburnum taxa: V. erosum, V. fengyangshanense, V. lancifolium, and V. squamulosum. Three plastid markers (matK, rbcL, and ndhF) were extracted from the chloroplast genomes using Geneious Prime software, while nrITS sequences were isolated and curated with ITSx. Although whole chloroplast genome data were generated, phylogenetic analyses focused on these three plastid loci combined with nrITS due to the limited availability of complete chloroplast genome sequences for Chinese Viburnum species. This marker combination strategically targets evolutionarily informative regions that are widely represented in existing datasets, enabling comprehensive phylogenetic comparisons across Chinese Viburnum lineages. (Line 110-127 in “Revised Manuscript with Track Changes.docx”)

Weakness 4: Limited comparison with prior phylogenetic studies

Response: The primary objective of this study is to elucidate the phylogenetic relationship betweenV. erosum and V. fengyangshanense through integrated morphological and molecular evidence. Our findings robustly support the monophyly of the Viburnum and the sect. Odontotinus (BS = 98, PP = 1.00). Notably, the samples of V. fengyangshanense form a well-supported clade (BS = 95, PP = 0.99) with two accessions of V. erosum, which providing conclusive evidence to resolve the phylogenetic placement of these taxa. These results not only achieve the primary aim of this study but also offer new insights into the evolutionary history of the Viburnum lineage in East Asia. (Line 162-171 in “Revised Manuscript with Track Changes.docx”)

---

## [Editor Report · Decision Letter 1]

20 Feb 2025

A new synonym for Viburnum erosum (Viburnaceae) in East China, based on morphological and molecular evidence

PONE-D-24-45964R1

Dear Dr. TANG,

We’re pleased to inform you that your manuscript has been judged scientifically suitable for publication and will be formally accepted for publication once it meets all outstanding technical requirements.

Kind regards,

Alia Ahmed

Academic Editor

PLOS ONE
---

## [Editor Report · Acceptance letter]

PONE-D-24-45964R1

PLOS ONE

Dear Dr. Tang,

I'm pleased to inform you that your manuscript has been deemed suitable for publication in PLOS ONE. Congratulations! Your manuscript is now being handed over to our production team.

Kind regards,

on behalf of

Dr. Alia Ahmed

Academic Editor

PLOS ONE